

# Ageing- and AAA-associated differentially expressed proteins identified by proteomic analysis in mice

Jinrui Ren[1,2], Jianqiang Wu[2,3], Xiaoyue Tang[3], Siliang Chen[1], Wei Wang[1], Yanze Lv[1], Lianglin Wu[1], Dan Yang[4] and Yuehong Zheng[1,2]

[1] Department of Vascular Surgery, Peking Union Medical College, Chinese Academy of Medical Sciences, Peking Union Medical College Hospital, Beijing, China
[2] State Key Laboratory of Complex Severe and Rare Disease, Peking Union Medical College Hospital, Chinese Academy of Medical Sciences and Peking Union Medical College, Beijing, China
[3] State Key Laboratory of Complex Severe and Rare Diseases, Medical Research Center, Peking Union Medical College Hospital, Chinese Academy of Medical Sciences and Peking Union Medical College, Beijing, China
[4] Department of Computational Biology and Bioinformatics, Institute of Medicinal Plant Development, Chinese Academy of Medical Sciences and Peking Union Medical College, Beijing, China

Corresponding author
Yuehong Zheng,
yuehongzheng@yahoo.com

## ABSTRACT

**Background:** Abdominal aortic aneurysm (AAA) is a disease of high prevalence in old age, and its incidence gradually increases with increasing age. There were few studies about differences in the circulatory system in the incidence of AAA, mainly because younger patients with AAA are fewer and more comorbid nonatherosclerotic factors.

**Method:** We induced AAA in ApoE$^{-/-}$ male mice of different ages (10 or 24 weeks) and obtained plasma samples. After the top 14 most abundant proteins were detected, the plasma was analyzed by a proteomic study using the data-dependent acquisition (DDA) technique. The proteomic results were compared between different groups to identify age-related differentially expressed proteins (DEPs) in the circulation that contribute to AAA formation. Gene Ontology (GO), Kyoto Encyclopedia of Genes and Genomes (KEGG), and protein–protein interaction (PPI) network analyses were performed by R software. The top 10 proteins were determined with the MCC method of Cytoscape, and transcription factor (TF) prediction of the DEPs was performed with iRegulon (Cytoscape).

**Results:** The aortic diameter fold increase was higher in the aged group than in the youth group ($p < 0.01$). Overall, 92 DEPs related to age and involved in AAA formation were identified. GO analysis of the DEPs showed enrichment of the terms wounding healing, response to oxidative stress, regulation of body fluid levels, ribose phosphate metabolic process, and blood coagulation. The KEGG pathway analysis showed enrichment of the terms platelet activation, complement and coagulation cascades, glycolysis/gluconeogenesis, carbon metabolism, biosynthesis of amino acids, and ECM-receptor interaction. The top 10 proteins were Tpi1, Eno1, Prdx1, Ppia, Prdx6, Vwf, Prdx2, Fga, Fgg, and Fgb, and the predicted TFs of these proteins were Nfe2, Srf, Epas1, Tbp, and Hoxc8.

**Conclusion:** The identified proteins related to age and involved in AAA formation were associated with the response to oxidative stress, coagulation and platelet activation, and complement and inflammation pathways, and the TFs of these proteins might be potential targets for AAA treatments. Further experimental and biological studies are needed to elucidate the role of these age-associated and AAA-related proteins in the progression of AAA.

## INTRODUCTION

Abdominal aortic aneurysms (AAAs) are the most common form of aortic aneurysm, and AAA rupture and associated catastrophic physiological damage result in an overall mortality rate of over 80%; morbidity has been shown in up to 8% of men aged >65 years (*George et al., 2015*; *Sakalihasan, Limet & Defawe, 2005*). The incidence rate of AAA in males aged >65 years increases by 40% every 5 years, indicating that age is a significant risk factor for AAA (*Chichester Aneurysm Screening Group et al., 2001*; *Nordon et al., 2011*). Histopathologically, AAA is related to increased inflammatory cell infiltration, abnormal oxidative stress, medial elastin degradation, and medial collagen deposition (*Daugherty & Cassis, 2002*; *Maegdefessel, Dalman & Tsao, 2014*). With age-related alterations, several alterations in the vasculature, such as enhanced inflammatory response, vascular stiffening, and oxidative stress, predispose aged arteries to vascular disease (*Bourantas et al., 2018*; *Dominic et al., 2020*). Furthermore, the relationship between AAA and aging is also reflected at the cellular level. Cellular senescence is a hallmark of aging and might play a vital role in the development of AAA, and this link has been shown in many previous studies. In intra-aortic walls, senescence of vascular mesenchymal stromal cells impairs their remodeling ability, which can support the formation and development of AAA, linking vascular senescence and inflammation (*Chen et al., 2016*; *Teti et al., 2021*). Moreover, the accumulation of senescent cells in the perivascular adipose tissue and abdominal aortic tissue promotes the formation of aneurysms through induction of inflammation, oxidative stress, and leukocyte adhesion (*Parvizi et al., 2021*).

In addition to the changes in the aortic wall and surrounding microenvironment that affect the formation of AAA, changes in the composition of the circulatory system, which result in high AAA morbidity in older people, might play an essential role in the formation of AAA. Increased insulin-like growth factor 1 (IGF1) expression and an increased IGF1/IGFBP3 ratio were reported to be associated with AAA, whereas IGFBP1 was independently associated with increased aortic diameter. Components of the IGF1 system may contribute to or be markers of aortic dilatation in older adults (*Yeap et al., 2012*). Elevated tHcy is associated with the presence of AAA in older men (*Wong et al., 2013*). Levels of the biomarkers C-reactive protein (CRP), proneurotensin (PNT), copeptin (CPT), lipoprotein-associated phospholipase 2 (Lp-PLA2), cystatin C (Cyst C), midregional proatrial natriuretic peptide (MR-proANP), and midregional

proadrenomedullin (MR-proADM) were not associated with the aortic diameter at ultrasound examination after 14–19 years of follow-up (*Taimour et al., 2017*). Given that AAA cases can be sporadic and/or associated with genetic diseases and infections, exploration of AAA models in different age groups is warranted.

Overall, it is still unclear why elderly patients (>65 years) are most likely to develop AAA, and the mechanisms for the significantly accelerated development and progression of AAA in these patients remain to be determined. We hypothesize that aging is associated with the formation of AAA, and there may be age-related changes in the circulatory system in addition to changes in the senescence of cells in the aortic wall. Therefore, we generated AAA models by using mice of different ages, and we examined the changes in their plasma using proteomic techniques to uncover age-related changes in the circulatory system that promote AAA formation.

## METHODS AND MATERIALS

### Construction of the AAA animal model and sample collection

For the mice model used in this study, ApoE knockout (KO) mice on a C57BL/6 background were obtained from the Beijing Vitalriver Laboratory Animal Co. (Beijing, China). All mice were raised in specific-pathogen-free conditions under 12/12 cycle of light at room temperature (24–27 °C) and allowed free access to food and water in a specific pathogen-free environment. The AAA model was generated by administering Ang II (1,000 ng/kg per minute; Sigma–Aldrich, St. Louis, MO, USA) or saline to male mice (10 weeks of age in the youth group or 24 weeks of age in the aged group) for 28 d *via* implantation of ALZET mini-osmotic pumps (model 2007; DURECT, Cupertino, CA, USA), as in our previous studies (*Ren et al., 2015*; *Zhang et al., 2020*). The mice were, mice were euthanized by cervical dislocation after inhalational anesthesia with 0.1% pentobarbital, and the blood of mice was collected by cardiac puncture at 16 weeks of age for the youth group (or at 28 weeks for the aged group) for further histological and molecular analyses. The diameter of the abdominal aorta in mice was measured by vernier caliper, and the diameter increased more than fifty percent, which was identified as aneurysm formation. All animal studies were approved by the Institutional Animal Care and Use Committee of Peking Union Medical College Hospital (JS-2629), and experiments conformed to the Guide for the Care and Use of Laboratory Animals (National Institutes of Health publication no. 85–23, 1996). The plasma was separated from the blood by centrifugation at 2,000 rpm/min for 10 min at 4 °C, and all samples were stored at −80 °C.

### Separation of the top 14 most abundant proteins

The top 14 most abundant proteins in plasma were detected with Thermo Scientific High-Selected Top 14 Abundant Protein Detection Resin (A36370; Thermo Fisher Scientific, Waltham, MA, USA). The depletion spin column was equilibrated to room temperature, the column screw cap was removed, and up to 10 µL of plasma was added directly to the resin slurry in the column. The column was capped and inverted several times until the resin solution was completely homogenous. The mixture was incubated in the column with gentle end-over-end mixing for 10 min at room temperature. The sample
was ensured to mix with the resin during incubation period. Alternatively, the sample was gently vortexed every few minutes. After incubation, the bottom closure was snapped off, and the top cap was loosened. The mini column was placed into a two mL collection tube and centrifuged at 1,000*g* for 2 min. The column containing the resin was discarded, and the filtrate contained the sample with albumin, IgG, and other abundant proteins removed. The BCA assay (23225; Thermo Fischer Scientific, Waltham, MA, USA) was employed to measure the protein concentration of the plasma.

## Protein digestion

The mouse plasma proteins were prepared using filter-aided sample preparation (FASP) methods (*Wisniewski et al., 2009*). The sample was deoxidized with 20 mM DTT (95 °C, 15 min) and alkylated with 50 mM IAA (room temperature in the dark, 45 min). Then, 6X precooled-acetone precipitation was used to extract proteins at −20 °C overnight. The proteins were redissolved with 20 mM Tris, loaded onto a 30 kDa cutoff ultrafiltration unit, and centrifuged at 14,000*g* at 4 °C. After being washed 3 times with 20 mM Tris, the samples were digested with MS-grade trypsin (1:50) at 37 °C overnight. After centrifugation at 4 °C for 10 min, the peptide solution was collected and stored at −80 °C for later use.

## LC–MS/MS

Peptides were separated with a capillary liquid chromatography (LC) column (75 μm × 500 mm, C18, 3 μm; Kyoto Monotech, Kyoto, Japan). The following 60-min gradient at a flow rate of 0.6 μL/min was used: 0–1 min, 6–11% solvent B (mobile phase A: 0.1% formic acid in $H_2O$; mobile phase B: 0.1% formic acid in acetonitrile); 1–9 min, 11–17% solvent B; 9–40 min, 17–29% solvent B; 40–50 min, 29–37% solvent B; 50–55 min, 37–100% solvent B; 55–60 min, 100% solvent B. The autosampler temperature was set at 4 °C, and the column was maintained at ambient temperature. An Orbitrap Q-Exactive HF (Thermo Scientific, Dreieich, Germany) mass spectrometer was employed to analyze the eluted peptides from LC. Data were acquired with dada-dependent acquisition (DDA) using the following parameters: data-dependent tandem mass spectrometry (MS/MS) scans per full scan in top-speed mode for 3 s; MS scan resolution of 120,000; MS/MS scan resolution of 30,000; 30% HCD collision energy; charge-state screening for +2 to +7; dynamic exclusion duration of 30 s; maximum injection time of 45 ms.

The raw MS data files were searched using SpectroMine (version 3.0; Biognosys, Schlieren, Switzerland) with Sequest HT against the SwissProt mouse database (http://www.UniProt.org) for quantitative and qualitative analysis. A maximum of two missed cleavages for trypsin was used; cysteine carbamidomethylation was set as a fixed modification, and oxidation (M), carbamylation, cysteine carbamidomethylation and deamidation were used as variable modifications. Protein identifications were required to have a false discovery rate (FDR) of less than 1.0% at the protein level with at least two unique peptides.

## Statistical analysis

Quantitative results are expressed as the mean± S.E.M. Comparisons of parameters between groups were made by t test. A value of $p < 0.05$ was considered to be statistically significant. Statistical significance was evaluated with R (4.1.1).

## DEP analysis

Pearson correlation tests were performed to evaluate the significance of differences in protein expression between groups with the DEP software package of R (4.1.1). After the data were standardized using the limma software package of R, we identified DEPs by comparing AAA with normal samples in the youth group (or aged group). Principal component analysis (PCA) was implemented using R. If the change factor was greater than 1.2-fold (|fold change| ≥ 1.2) and the $p$ value was ≤ 0.05, a protein was considered to be differentially expressed. A volcano plot, heatmap, and Venn diagram of the DEPs were generated using the ggplot2, pheatmap, and Venn software packages of R, respectively.

## GO, KEGG, and MeSH enrichment analyses

Gene Ontology (GO), Kyoto Encyclopedia of Genes and Genomes (KEGG), and MeSH enrichment analyses were performed using the clusterProfiler package (*Yu et al., 2012*) in R (4.1.1). A hypergeometric distribution was used to analyze and calculate the significance levels of the significantly affected signaling pathways of the DEPs ($p < 0.05$).

## PPI network and TF analyses

The STRING database was used to construct a protein–protein interaction (PPI) network of the aging-related DEPs. The PPI file was imported into Cytoscape (3.6.5). The degree, closeness, intermediate degree of each node in the network, and average value of each protein's nodal degree were defined as the threshold of the PPI network nodes, and the proteins whose degrees were more significant than the threshold value were selected. The key nodes (the hub proteins) of the PPI network were identified by the MCC of cytoHubba and mapped, and the correlation scores of the nodes and their interacting proteins were calculated. The functions of the hub proteins were predicted by the GeneMANIA app of Cytoscape. The transcription factors (TFs) of the top 10 hub proteins were predicted by iRegulon (*Chen et al., 2016*).

# RESULTS

## Workflow of the proteome analysis

In this study, 20 samples were enrolled and divided into four groups: the normal (youth) group included mice of ten weeks of age that received saline administration for 28 days ($n = 5$); the AAA (youth) group included mice of ten weeks of age that received Ang II administration for 28 days ($n = 5$); the normal (aged) group included mice of 24 weeks of age that received saline administration for 28 days ($n = 5$); and the AAA (aged) group included mice of 24 weeks of age that received Ang II administration for 28 days ($n = 5$). Fig. 1A shows the general workflow of this study. By measuring the diameter of the abdominal aorta in mice, it was found that the AAA/normal abdominal aortic diameter

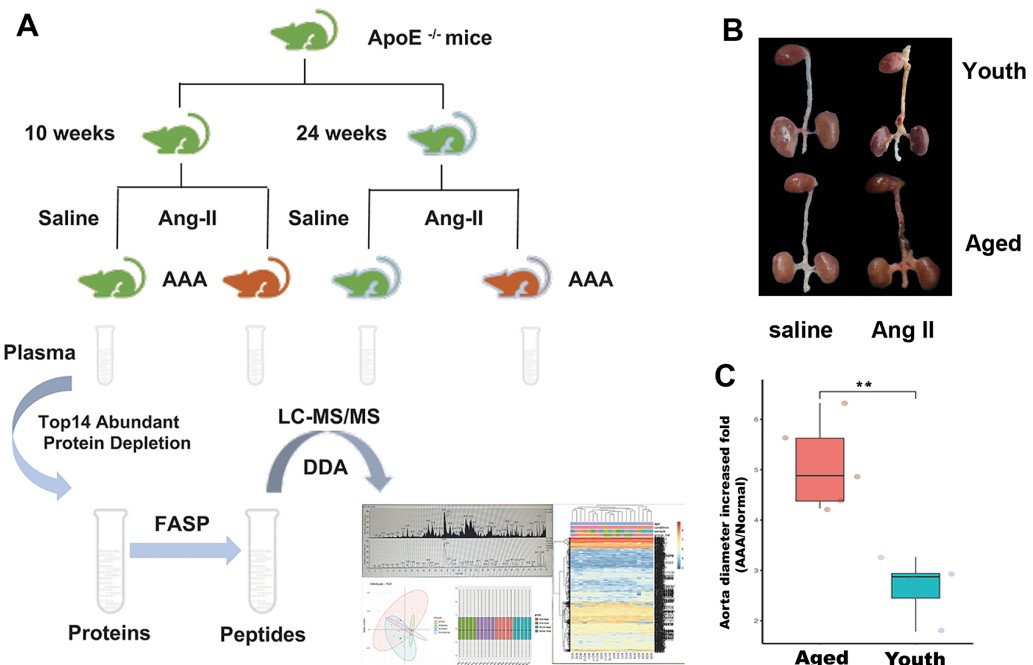

**Figure 1 The workflow of this study and the aortic images in different groups.** The workflow of this study (A) and the aortic images in different groups (B) the expanded fold of the abdominal aorta is higher in the aged group compared with the youth group (C, **$p < 0.01$).

increase (as represented by the AAA to normal ratio) in the elderly group was significantly greater than that in the young group ($p < 0.01$, Figs. 1B and 1C).

All of the samples were assessed in two technical replicates by DDA 2D-LC/MS analysis, and the mean value of two test results was taken for each sample. In the DDA analysis, 992 proteins were detected with a protein FDR <1%. For each sample, 700 protein groups were identified by peptide-spectrum matching. Pearson correlation analysis showed a major cluster of samples in relevant groups (Fig. 2A). In total, 535 protein groups with quantitative data in more than 70% of samples in each group were selected for further analysis (Fig. 2B). Sample normalization was good, and intergroup grouping was relatively straightforward (Figs. 2C and 2D).

## DEP analysis

There were 70 DEPs between the AAA (youth) and normal (youth) groups. Thirty-three of them were upregulated, and 37 were downregulated in the AAA (youth) group compared to the normal (youth) group (Figs. 3A and 3B). In the AAA (aged) group compared with the normal (aged) group, 108 DEPs were identified, 67 of which were upregulated and 41 of which were downregulated in the AAA (aged) group (Figs. 3C and 3D). Moreover, we found that 138 DEPs were significantly different between the AAA (aged) group and the AAA (youth) group. Moreover, 83 of the DEPs were upregulated, while 55 were downregulated in the AAA (aged) group (Figs. 3E and 3F).

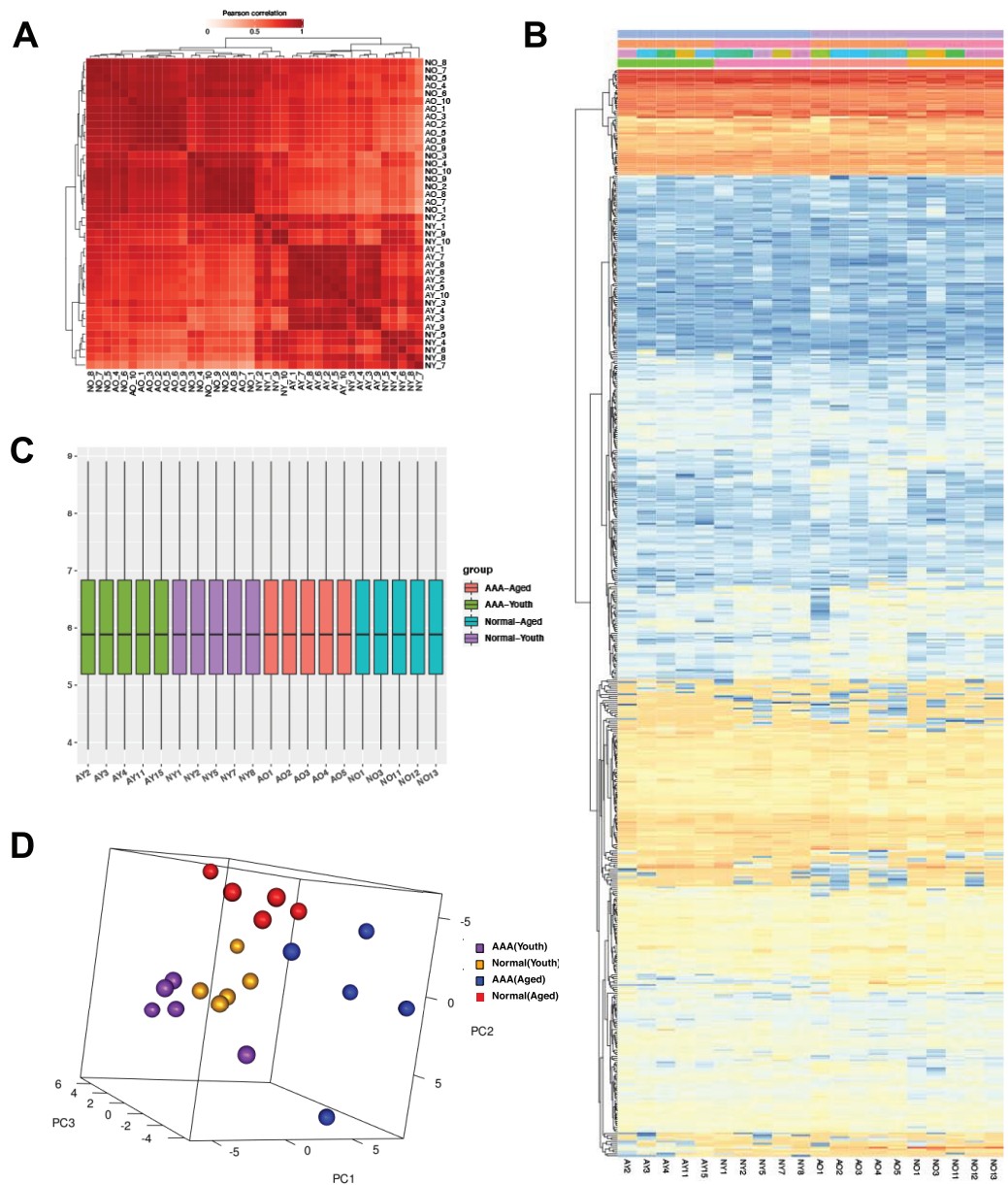

**Figure 2 The Pearson correlated test in samples that underwent twice technically repeating (A). Unsupervised hierarchical clustering of plasma proteome samples shows clustering of samples (columns) according to protein expression profiles (rows) (B). The proteins expression conditions after standardized normalizer (C) and Principal Component Analysis show the samples' distinction according to groups (D).**

There were three independent DEP sets (Fig. 3G): AAA (youth) *vs* normal (youth) DEPs, AAA (aged) *vs* normal (aged) DEPs, and AAA (aged) *vs* AAA (youth) DEPs. In total, 92 DEPs that might be related to the aging progress and the formation of AAA were identified in the AAA (aged) *vs* AAA (youth) DEP set and the AAA (youth) *vs* normal

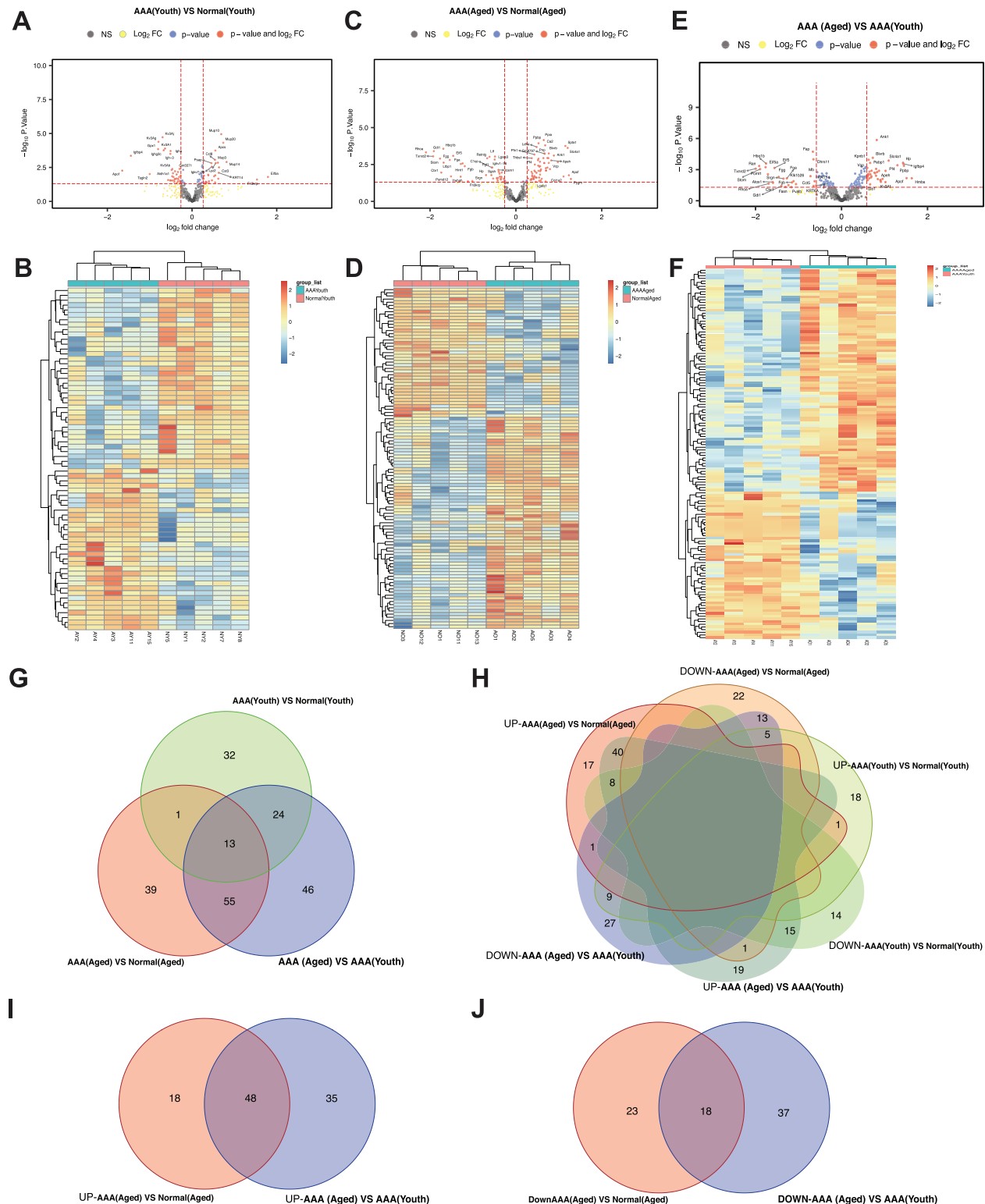

**Figure 3 Differential expressed proteins analysis.** The X-axis represents log2Fold Change, and Y-axis represents −log10 (*p*.Value). Red dots represent proteins with |foldchange| > 1.2 and −log(*p*.Value) < 0.05. (A) AAA (Youth) *vs* Normal (Youth); (C) AAA (Aged) *vs* Normal (Aged); (E) AAA (Aged) *vs* AAA (Youth). Heatmap of DEPs and the diagram presents the result of a two-way hierarchical clustering of all the DEPs and samples (B), AAA (Youth) *vs* Normal (Youth); (D) AAA (Aged) *vs* Normal (Aged); (F) AAA (Aged) *vs* AAA (Youth). The Veen diagrams show overlap

**Figure 3** (continued)
between DEPs in different ways of comparison (G). The Veen diagrams show overlap between up-regulated DEPs and down-regulated DEPs in different ways of comparison (H). The up-regulated VEEN graph showed 48 DEPs had similarly down-related expression trends in 2 DEPs sets (I), AAA (Aged) *vs* Normal (Aged); AAA (Aged) *vs* AAA (Youth). The down-regulated VEEN graph showed 18 DEPs had similarly down-related expression trends in 2 DEPs sets (J) AAA (Aged) *vs* Normal (Aged); AAA (Aged) *vs* AAA (Youth).     

(youth) DEP set (or the AAA (aged) *vs* normal (aged) DEP set) (Table 1). The DEPs were also divided and plotted in a Venn diagram in six differential subsets based on whether they were upregulated or downregulated in the sets (Fig. 3H). To further analyze the expression of DEPs in different sets, we generated Venn diagrams of upregulated and downregulated proteins. The upregulated Venn diagram showed 48 common DEPs in both the AAA (aged) *vs* normal (aged) and AAA (aged) *vs* AAA (youth) sets (Fig. 3I). The downregulated Venn diagram showed 18 common DEPs in both the AAA (aged) *vs* normal (aged) and AAA (aged) *vs* AAA (youth) sets (Fig. 3J).

## Functional annotation of the DEPs

To explore the molecular mechanisms and biological progress of the DEPs, which might be related to aging and AAA formation, 89 proteins of the 92 DEPs (as 3 DEPs failed to match with Entrez ID proteins) were subjected to GO, KEGG, and MeSH annotation analyses. The results of the GO and KEGG pathway for DEPs are shown in Tables S1 and S2. The GO biological process (*Sohrabpour, Kearns & Massari, 2016*) analysis showed that the DEPs were associated with blood coagulation, hemostasis, coagulation, and hydrogen peroxide catabolic processes; the GO cellular component (CC) analysis showed that the DEPs were related to the cell cortex, platelet alpha granules, the myelin sheath, and blood microparticles; the GO molecular function (MF) analysis indicated the DEPs were related to antioxidant activity, peroxidase activity and oxidoreductase activity (Fig. 4A). The top five enriched GO terms were wound healing, response to oxidative stress, regulation of body fluid levels, ribose phosphate metabolic process and blood coagulation (Fig. 4B, Table 1). The primary enriched GO process terms were coagulation, hemostasis, and hydrogen peroxide catabolic process, which were associated with Gpx1, Prdx1, Prdx2, and Prdx6 (Fig. 4C). The DEPs showed enrichment of the KEGG pathway terms platelet activation, complement and coagulated cascades, glycolysis/gluconeogenesis, carbon metabolism, biosynthesis of amino acids, and ECM-receptor interaction (Figs. 4D and 4E, Table 1). The MeSH analysis of the DEPs showed annotation of the terms thrombosis, leukemia, lung diseases, anemia, and malaria (Fig. 4F).

## Identification of hub proteins of the DEPs

The DEPs were mapped into a coexpression network obtained from previous protocols using STRING and Cytoscape. The weighted edge threshold was set as 71 nodes, and 358 edges were in the DEP coexpression network (Figs. 5A, Tables S3). The DEPs were clustered into six main clusters according to their function using STRING (MCL clusters) and mapped through Cytoscape (Fig. 5B). The most significant cluster of the DEP coexpression network detected by MCC (Cytoscape, CytoHubba) consisted of 10 genes (Fig. 5C, Table 2). Triosephosphate isomerase 1 (Tpi1, which belongs to the

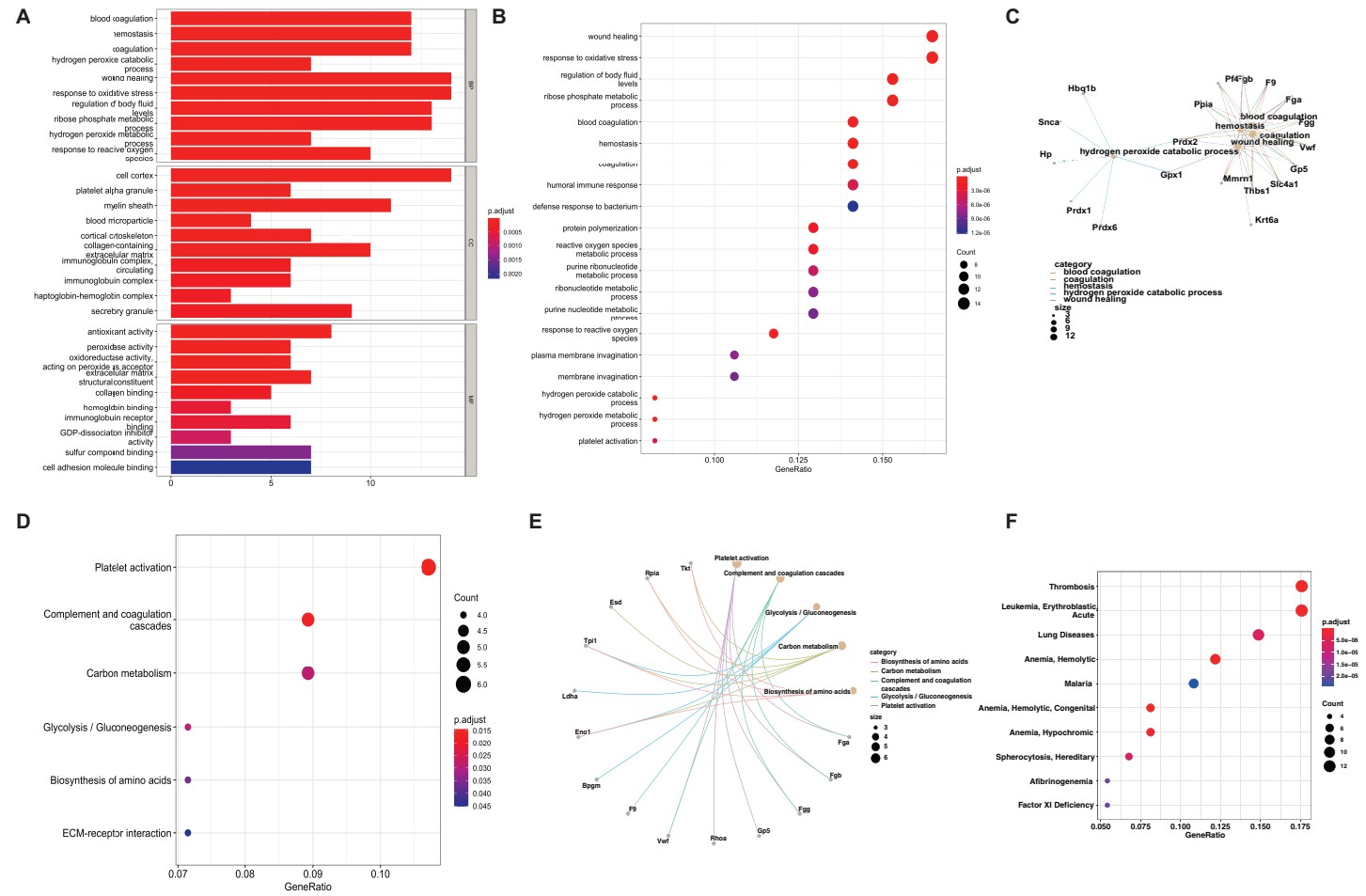

**Figure 4 The significantly enriched GO, KEGG pathways, and Mesh words an analysis of the 92 DEPs which are age-related involved in AAA formation.** The GO analysis on CC, BP, and MF (A), top 20 enriched GO progress of 92 DEPs (B), and top five enriched GO progresses with their related proteins were mapped (C). The KEGG pathways of 92 DEPs (D) and top five KEGG terms with their related proteins were mapped (E). The Mesh analysis of these DEPs, $p < 0.05$ (F).

**Table 1 GO and KEGG pathways terms with top 5 count number of DEPs.**

| ID | Terms | Count | *p* value |
|---|---|---|---|
| **GO** | | | |
| GO:0042060 | Wound healing | 14 | 4.21E–11 |
| GO:0006979 | Response to oxidative stress | 14 | 2.59E–10 |
| GO:0005938 | Cell cortex | 14 | 7.03E–12 |
| GO:0050878 | Regulation of body fluid levels | 13 | 6.08E–10 |
| GO:0019693 | Ribose phosphate metabolic process | 13 | 7.19E–10 |
| **KEGG** | | | |
| mmu04611 | Platelet activation | 6 | 0.00011651 |
| mmu04610 | Complement and coagulation cascades | 5 | 0.00027545 |
| mmu01200 | Carbon metabolism | 5 | 0.0009225 |
| mmu00010 | Glycolysis/Gluconeogenesis | 4 | 0.00079126 |
| mmu01230 | Biosynthesis of amino acids | 4 | 0.00146719 |

**Table 2 Hub proteins of DEPs were age-related and involved in AAA formation.**

| Rank | Name | ENTREZID | Official full gene name | AAA (Aged) *vs* Normal (Aged) | AAA (Aged) *vs* AAA (Youth) | AAA (Youth) *vs* Normal (Youth) |
|------|------|----------|-------------------------|---------------------------------|------------------------------|----------------------------------|
| 1 | Tpi1 | 21991 | Triosephosphate isomerase | + | + | n |
| 2 | Eno1 | 13806 | Alpha-enolase | + | + | n |
| 3 | Prdx1 | 18477 | Peroxiredoxi 1 | + | + | n |
| 4 | Ppia | 268373 | Peptidyl-prolyl cis-trans isomerase A | + | + | n |
| 5 | Prdx6 | 11758 | Peroxiredoxin 6 | + | + | n |
| 6 | Vwf | 22371 | Von Willebrand factor | + | + | n |
| 7 | Prdx2 | 21672 | Peroxiredoxin 2 | + | + | − |
| 8 | Fga | 14161 | Fibrinogen alpha | − | − | n |
| 9 | Fgg | 99571 | Fibrinogen beta | − | − | n |
| 9 | Fgb | 110135 | Fibrinogen gamma | − | − | n |

Notes:
+ up-regulated expression.
− down-regulated expression.
n none differential expression.

triosephosphate isomerase family), alpha-enolase (Eno1, a multifunctional enzyme that, in addition to its role in glycolysis, plays a part in various processes, such as growth control, hypoxia tolerance, and allergic responses), and peroxiredoxin-1 (Prdx1, a protein that plays a role in cell protection against oxidative stress) were the proteins with the top three degrees in the cluster (Fig. 5C). We also analyzed the functions and networks of these hub proteins through GeneMANIA, and the main functions were peroxidase activity, purine ribonucleoside diphosphate metabolic processes, ADP metabolic processes, negative regulation of response to wounding, blood coagulation, endothelial cell apoptotic processes, and negative regulation of the extrinsic apoptotic signaling pathway (Fig. 5D). The TFs of the top 10 hub proteins were predicted by iRegulon, and the top 5 TFs were Nfe2, Srf, Epas1, Tbp, and Hoxc8 (Fig. 5E, Table 3).

# DISCUSSION

We conducted a proteome analysis of plasma from induced AAA mice and normal mice of different ages (youth group and aged group). Ninety-two DEPs in plasma were identified to be associated with aging and AAA formation. The GO analysis of these DEPs showed enrichment of the terms wound healing, response to oxidative stress, regulation of body fluid levels, ribose phosphate metabolic process and coagulation, processes that might be associated with the aging process and contribute to the formation of AAA. The main GO enrichment terms were wounding healing, response to oxidative stress, regulation of body fluid levels, ribose phosphate metabolic process and blood coagulation. The DEPs were associated with the KEGG pathway terms platelet activation, complement and coagulated cascades, glycolysis/gluconeogenesis, carbon metabolism, biosynthesis of amino acids, and ECM-receptor interaction. The hub protein cluster was the most significant in the DEP coexpression network selected by cytoHubba. The top 10 proteins in the cluster were Tpi1,

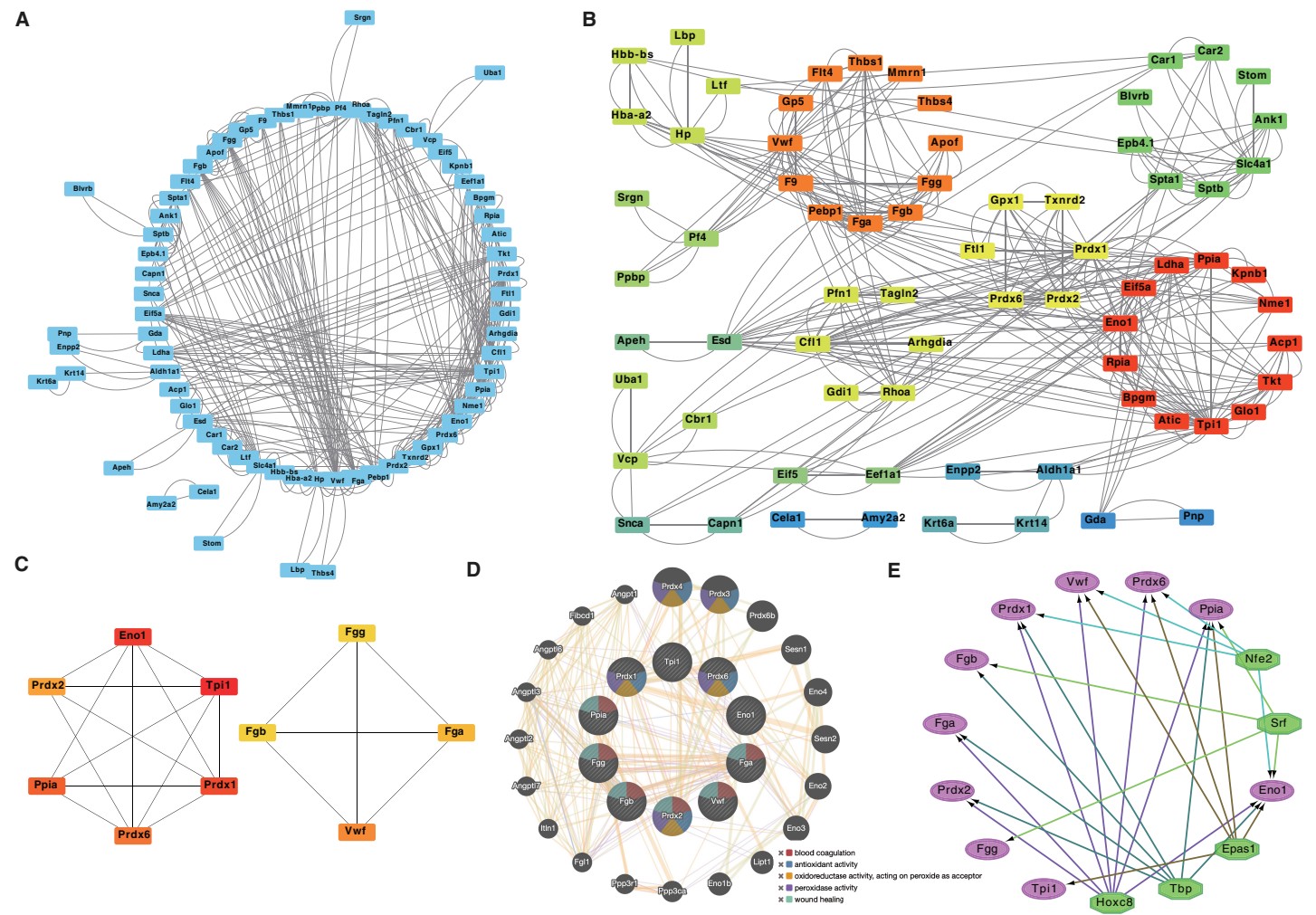

**Figure 5 DEPs co-expression network, hub proteins cluster, and transcription factors predict.** DEPs co-expression network (A) and protein function cluster by String (MCL cluster function) (B). Hub proteins cluster of DEPs co-expression network (C). Hub proteins functions and networks (D). The predicting transcription factors of hub proteins (E).

**Table 3 The predicting transcription factors of Hub proteins.**

| TF | Uniport id | Full names |
| --- | --- | --- |
| Hoxc8 | P09025 | Homeobox protein Hox-C8 |
| Tbp | P62340 | TATA box-binding protein-like 1 |
| Epas1 | P97481 | Endothelial PAS domain-containing protein 1 |
| Srf | Q9JM73 | Serum response factor |
| Nfe2 | Q07279 | Transcription factor NF-E2 45 kDa subunit |

Eno1, Prdx1, Ppia, Prdx6, Vwf, Prdx2, Fga, Fgg, and Fgb. These proteins are potentially crucial proteins that are related to the aging process and the pathogenesis of AAA.

Arterial senescence may be associated with decreased vasodilation due to increased reactive oxygen species (ROS) production, whereas endothelial cell activation induces

procoagulant changes, which are also related to the inflammation and oxidative stress of AAA (*Sanchez-Infantes et al., 2021*; *Usui et al., 2015*). ROS are produced in large amounts during inflammation and can stimulate connective tissue-degrading proteases and smooth muscle cell (SMC) apoptosis, and ROS are increased locally in AAA and lead to enhanced oxidative stress (*Miller et al., 2002*). ROS control extracellular matrix degradation and remodeling by upregulating proteolytic enzymes, such as MMPs, with concurrent significant elevations in oxidative stress and proteolytic enzyme expression within aneurysmal walls covered by thin thrombi (</= 10 mm) compared with thick thrombus-covered walls (>/= 25 mm) (*Wiernicki et al., 2019*). In our study, oxidative stress was closely related to the occurrence of AAA and aging, and we found that Prdx1, 2 and 6 also contributed to ageing and AAA formation. Prdx proteins are a ubiquitous thiol-specific antioxidant enzymes that controls the level of intracellular peroxides and are involved in oxidative stress and signal transduction; in addition, the mammalian Prdx family includes six members: Prdx1-6 (*Rhee, Chae & Kim, 2005*; *Wood et al., 2003*). Simvastatin may inhibit NF-κB activity by inhibiting free radicals and TNF-α production, improving human AAA wall tissue (*Piechota-Polanczyk et al., 2012*). Elevated Prdx-1 levels in the serum and tissues of AAA patients were associated with AAA size and the AAA growth rate (*Martinez-Pinna et al., 2011*; *Ramella et al., 2018*; *Rasiova et al., 2019*). Prdx2 plays a role as a negative regulator of the pathological process of AAA (*Jeong et al., 2020*; *Martinez-Pinna et al., 2014*). However, in other studies, Prdx2 was found to be an inducer of ruptured AAA (*Urbonavicius et al., 2009*), while Prdx6 expression was elevated in patients with AAA; in addition, in AAA tissue, prdx6 colocalized with neutrophils, vascular SMCs, and oxidized lipids. Prdx6 is elevated in AAA plasma, reflecting increased systemic oxidative stress, and there is a positive correlation between prdx6 levels and AAA diameter (*Burillo et al., 2016*). Our study found that Prdx2 was downregulated in the AAA (youth) group compared with the normal (youth) group. Furthermore, it was upregulated in the AAA (aged) group compared with the other groups, which indicates that Prdx2 might promote AAA formation in aged people, but more studies are needed.

Metabolic activities control cell fate, and glucose is the primary energy source for metabolic activities and an essential substrate for protein and lipid synthesis in mammalian cells. However, Increased glycolysis contributes to the pathogenesis of various diseases, such as cancer, degenerative diseases, metabolic syndrome, and infection (*Afonso et al., 2020*; *Kolovou, Kolovou & Mavrogeni, 2014*). Increased glycolysis is associated with many inflammatory processes and contributes to AAA formation, and macrophages produce inflammasomes (*Sun et al., 2015*). Recent studies have shown that the modified glucose analog 18-fluorodeoxyglucose (18F-FDG) accumulates in atherosclerosis-based AAA, suggesting an association between inflammation and glycolysis in the pathogenesis of AAA development. Intraperitoneal injection of glycolysis inhibitors and 2-deoxyglucose significantly attenuated aortic aneurysm formation in male C57BL/6J mice with CaCl$_2$-induced abdominal aorta dilatation or male ApoE KO mice with decreased angiotensin II infusion. Increased glycolysis activity in the aortic wall contributes to the pathogenesis of aneurysm development (*Tsuruda et al., 2012*). Rescue intervention with the glycolytic inhibitor pfk15 in an AngII model showed that interfering with the glycolytic

switch prevented aneurysm formation (*Gabel et al., 2021*). Triosephosphate isomerase (Tpi1) can catalyze the switch between glycolysis and gluconeogenesis in the presence of dihydroxyacetone and 3-D-glyceraldehyde (*Rodriguez-Almazan et al., 2008*). Tpi1, a hub protein in our study, has been identified as a human ageing-related protein (*de la Mora-de la Mora et al., 2015*), but there has been less research in AAA. Alpha-enolase (Eno1) is a glycolytic enzyme that catalyzes the conversion of 2-phosphoglycerate into phosphoenolpyruvate, and it is involved in various processes, such as growth control, hypoxia tolerance and anaphylaxis (*Huang et al., 2018*). Eno1 may also function in the intravascular and pericellular fibrinolytic system due to its ability to act as a receptor and activator of plasminogen on the surface of several cell types, such as leukocytes and neurons. Eno1 is a significant facilitator of plasminogen activation on the leukocyte surface (*Lopez-Alemany et al., 2003*; *Sugahara et al., 1992*). Therefore, the roles of Tpi1 and Eno1 in the mechanism by which glycolysis contributes to AAA should be investigated.

In our study, the age-related DEPs in Ang II-induced AAA models involved in AAA formation were found to be significantly enriched in the coagulation and platelet activation pathways. Coagulation and platelet activation play a crucial role in the occurrence, progression, and rupture of AAA. By detecting the plasma concentrations of thrombin−antithrombin (TAT) complexes, platelet factor 4 (PF4), and D-dimers in AAA patients, it was found that an increase in D-dimers and TAT complex levels can predict progression of the disease and the growth of aneurysms in patients with AAA or subaortic dilatation (*Sundermann et al., 2018*). Inhibition of FXa/FIIa can downregulate the activation of the Smad2/3 signaling pathway and MMP2 expression mediated by protease-activated receptor 2 (PAR2) to limit the severity of aortic aneurysm and atherosclerosis (*Moran et al., 2017*). Intraluminal thrombus (ILT) is a common pathological condition in AAA and is closely related to the formation and expansion of aneurysms. Some studies have suggested that ILT can promote the progression of AAA (*Michel et al., 2011*). ILT seems to have a biomechanical function by reducing peak wall stress, but it also has biological activity. In addition to increasing adventitia inflammation, ILT may enhance the degradation of the aortic wall. With the prominent accumulation of leukocytes in the ILT layer of the cavity, red blood cells and platelets release oxidative and proteolytic molecules that promote AAA wall destruction. In addition, they secrete chemotaxis-related molecules, which may increase the number and type of recruited cells (*Cameron, Russell & Owens, 2018*). Platelet activation and thrombus regeneration are key to the development of endovascular thrombosis and AAA. Inhibition of platelet activation limits the biological activity of thrombi in the cavity, thereby inhibiting the development of aneurysms (*Dai et al., 2009*). Platelet aggregation may be responsible for thrombus turnover in AAA, with luminal thrombi releasing markers of platelet activation that can be readily detected in plasma (*Touat et al., 2006*). The increased platelet numbers with age and increased risk of arterial thrombosis may be related to the increased interactions between endothelial cell-attached leukocytes and platelets. Antithrombotic drugs, such as low-molecular-weight heparin, rivaroxaban, and other anticoagulation drugs, also inhibit the formation of abdominal aortic aneurysms (*Grzela et al., 2008*).

Von Willebrand factor (VWF) is a multisubunit protein that anchors platelets to subendothelial collagen. It serves as a carrier protein for the VIII factor in plasma, and endothelial cell release of VWF plays an essential role in the formation of platelet aggregates and thrombosis and is closely associated with aging (*Alavi, Rathod & Jahroudi, 2021*). We found that VWF was upregulated in the AAA (aged) group (compared with the normal (aged) group or the AAA (aged) group), suggesting that it might be associated with aging and AAA formation. In previous studies, VWF was found to be expressed in aneurysm tissue and colocalized with hepatocyte growth factor (HGF) in the most severe part of the aneurysm-injured wall (*Shintani et al., 2011*). As age increases, VWF levels increase significantly, and thrombosis events increase significantly (*Alavi, Rathod & Jahroudi, 2021*). Fibrinogen is a vital protein involved in the processes of coagulation and hemostasis and plays a crucial role in the process of AAA formation, and fibrinogen has been proven to be a plasma biomarker of AAA formation and progression (*Sundermann et al., 2018*). A specific anti-fibrinogen antibody can induce AAA in mice through complement lectin pathway activation, and circulating antibodies in a subset of AAA patients react against fibrinogen or fibrinogen-associated epitopes in human aneurysmal tissues (*Zhou et al., 2013*). However, in our study, we found that fibrinogen alpha (Fga), fibrinogen beta (Fgb), and fibrinogen gamma (Fgg), which are the subunits of fibrinogen, were downregulated in the AAA (aged) group (compared with the normal (aged) group or the AAA (aged) group). However, they were not differentially expressed between the AAA (youth) group and the normal (youth) group. Thus, coagulation, platelet activation and ILT affect AAA formation through inflammation and ROS, which might be highly related to Vwf, Fga, Fgb and Fgg.

The complement system is a part of innate immunity that has been proven to play a critical role in many inflammatory diseases and plays an essential role in the pathogenesis of AAA (*Wu et al., 2010*). The age-related DEPs in plasma that were associated with AAA formation in this study showed similar enrichment of complement activation cascade. The innate immune response to autoantigens activates the complement system and activates the inflammatory cascade in AAA, and the activation of the complement system promotes AAA (*Zhou et al., 2012*). Alternative complement pathway involvement is not limited to this experimental model but is also evident in human AAA (*Pagano et al., 2009*). Compared with those in healthy controls, the serum C5a levels in AAA patients were significantly increased, and the serum C5a level was also significantly correlated with the maximal AAA diameter (*Zagrapan et al., 2021*). The complement system is a crucial mechanism involved in vascular remodeling and plays an essential role in the pathogenesis of AAA (*Martin-Ventura et al., 2019*; *Wu et al., 2010*). The alternative complement pathway also plays a vital role in the development of AAA: natural IgG antibodies direct alternative pathway-mediated complement activation, and C3 deposition is present in elastase-perfused aortic walls (*Zhou et al., 2012*).

Peptidyl-prolyl cis-trans isomerase A (Ppia) catalyzes the cis-trans isomerization of proline imine peptide bonds in oligopeptides, exerts strong chemotactic effects on leukocytes, activates endothelial cells in a proinflammatory manner, promotes endothelial cell chemotaxis and apoptosis, and plays a vital role in platelet activation and aggregation

(*Song et al., 2011*; *Xie, Li & Ge, 2019*). Cyclophilin A (CyPA; encoded by PPIA) is a ubiquitously expressed protein secreted in response to inflammatory stimuli that stimulates vascular SMC migration and proliferation, endothelial cell adhesion molecule expression, and inflammatory cell chemotaxis, thereby promoting atherosclerosis (*Nigro et al., 2011*). CyPA is highly expressed in vascular SMCs, secreted in response to ROS, and promotes inflammation, and ApoE$^{-/-}$Ppia$^{-/-}$ mice are entirely protected from AngII-induced AAA formation because they have reduced inflammatory cytokine expression, elastic lamina degradation, and aortic dilation (*Satoh et al., 2009*).

Our study predicted Nfe2, Srf, Epas1, Tbp, and Hoxc8 as potential TFs of the hub genes. Nuclear factor erythroid 2 (Nfe2) is a TF that plays a vital role in the response to oxidative stress and can bind to antioxidant response elements present in the promoter regions of many cytoprotective genes and promote their expression, thereby functioning as an antioxidant response protein (*Chaohui et al., 2018*; *Tan & de Haan, 2014*). Serum response factor (Srf) is a mouse TF required for cardiac differentiation and maturation and is involved in regulating atherosclerotic disease processes in vascular SMCs. Endothetal Per-Arnt-Sim domain protein 1 (Epas1) can regulate VEGF expression and appears to be involved in the development of the vascular and pulmonary tubular systems (*Stavik et al., 2016*; *Yang et al., 2021*). TATA-box-binding protein (Tbp) is a general TF that functions at the core of the DNA-binding multiprotein factor TFIID, which is also a crucial factor in the perivascular adipose tissue of AAA (*Piacentini, Chiesa & Colombo, 2020*).
The homeobox protein Hox-C8 (Hoxc8), a sequence-specific TF that is part of a developmental regulatory system that provides cells with specific positional identities along the anteroposterior axis and is related to proinflammatory and immunological genes and pathways in coronary artery disease (*Tan et al., 2020*). Although these TFs have not been confirmed to contribute to AAA formation (except Tbp), they have been shown to play a crucial role in vascular diseases in previous studies. Hence, these TFs might be potential regulators and treatment targets of AAA.

In this study, the results suggested that there is a quantity of circulation differences between aged and youth AAA mice models, which are age-related proteins involved in AAA formation. There are several limitations of this study. Firstly, this is a single animal experiment which had not been validated on human study. Secondly, the results lack of further validation with some other experimental techniques or a large sample. Nevertheless, this study is an exploratory study in which we utilized proteomics to attempt to discovery novel circulation varies, which might contribute to AAA formation in aged people. In the future, to verify the clinical application of these proteins, a large-sample, multicenter, prospective clinical validation study is needed.

## CONCLUSIONS

Our present study revealed circulatory environment changes that were associated with age and involved in AAA formation, and these changes were related to the response to oxidative stress, coagulation and platelet activation, complement, and inflammation. Prdx1, Prdx6, Tpi1, Ppia, Eno1, and Vwf might be circulatory factors that induce AAA in aged people, while Prdx2, Ffa, Ffb, and Ffg were differentially expressed based on

comparisons with other studies. Nfe2, Srf, Epas1, Tbp, and Hoxc8 might be crucial TFs that influence AAA formation in the circulatory system of aged people. These circulatory system proteins and TFs might be treatment targets or predictors of AAA, but more research is needed.

### Funding
This work was supported by grant from the Natural Science Foundation of China (grant number 82070492). The funders had no role in study design, data collection and analysis, decision to publish, or preparation of the manuscript.

### Grant Disclosures
The following grant information was disclosed by the authors:
Natural Science Foundation of China: 82070492.

### Competing Interests
The authors declare that they have no competing interests.

### Author Contributions

- Jinrui Ren conceived and designed the experiments, performed the experiments, analyzed the data, prepared figures and/or tables, authored or reviewed drafts of the paper, and approved the final draft.
- Jianqiang Wu conceived and designed the experiments, performed the experiments, authored or reviewed drafts of the paper, and approved the final draft.
- Xiaoyue Tang analyzed the data, prepared figures and/or tables, and approved the final draft.
- Siliang Chen analyzed the data, prepared figures and/or tables, and approved the final draft.
- Wei Wang performed the experiments, prepared figures and/or tables, and approved the final draft.
- Yanze Lv performed the experiments, prepared figures and/or tables, and approved the final draft.
- Lianglin Wu performed the experiments, analyzed the data, prepared figures and/or tables, and approved the final draft.
- Dan Yang conceived and designed the experiments, prepared figures and/or tables, authored or reviewed drafts of the paper, and approved the final draft.
- Yuehong Zheng conceived and designed the experiments, authored or reviewed drafts of the paper, and approved the final draft.

### Animal Ethics
The following information was supplied relating to ethical approvals (*i.e.*, approving body and any reference numbers):

All animal studies were approved by the Institutional Animal Care and Use Committee of Peking Union Medical College Hospital (JS-2629).

## Data Availability

The data is available at ProteomeXchange: PXD033725.

## Supplemental Information

Supplemental information for this article can be found online at http://dx.doi.org/10.7717/peerj.13129#supplemental-information.

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
