# Peer review of "Ageing- and AAA-associated differentially expressed proteins identified by proteomic analysis in mice"

_PeerJ, doi:10.7717/peerj.13129_

## Round 0.1 · original submission · Minor Revisions

The authors can follow the Reviewers' suggestions to improve the manuscript.

·

Basic reporting

The manuscript entitled “Ageing- and AAA-associated differentially expressed proteins identified by Proteomic Analysis” by Jinrui Ren et al. investigated the proteomic profiling of abdominal aortic aneurysm (AAA) in ApoE-/- male mice of different ages (10 or 24 weeks). They identified age- and AAA-related proteins were associated with the response to oxidative stress, coagulation and platelet activation, and complement and inflammation pathways.

Experimental design

None.

Validity of the findings

Authors identified AAA-related protein profile in ApoE-/- male mice model. How similar is the profile to human AAA patients?

Additional comments

1. Quality of Figure 2, 3, 4 and 5 could be improved. Do authors have images with higher resolution?
2. Figures should be referred in the same order as presented. Figure 2A should be referred before Figure 2B. Figure 3B and 3C should be referred before Figure 3D.
3. Table 1 showed GO and KEGG pathways terms with top 5 count number of DEPs, rather than 92 DEPs as stated in text line 221-224.
4. Were Figure 4 and Figure 5 switched? Figure 4 showed DEPs co-expression network, hub proteins cluster, and transcription factors predict and Figure 5 showed the significantly enriched GO, KEGG pathways, and Mesh words, not as stated in the manuscript.
5. Authors need to cite literature for roles of Tpi1, Eno1,and Prdx1 in text line 257-261.

·

Basic reporting

This manuscript by Jinrui Ren et al. contains some interesting data, and breaks some new ground. The concept that aging related proteins in circulatory may contribute to the development of AAA and this could have high physiological significance. To the authors credit, the data are in general supportive of the major conclusions. However, there appear to be some mistakes and flaws in the illustration and interpretations of some of the data and there is limited evidence for the overall concepts provided at this stage in the study.

Experimental design

The experiments design is very interesting to compare the different ages on the development of AAA.

Validity of the findings

1. 1B shows Ang II causes significant increases in Aorta diameter in both ages, while the combination of age and Ang II have profound effects. However, do authors know how many the higher expressed proteins from AAA groups were secreted proteins? Did any published data indicate where these proteins coming from?

2. In table 2, mistake in Fgb protein Rank number, should be number 10. Please correct it. In addition, Figure 1B showed the AngII woks in AAA youth mice, but authors did not show Hub proteins in AAA youth lab. Did authors compared the AAA youth VS Normal youth alone in Hub Proteins? Are there interesting proteins?

3. Rows 310-312, the authors wrote ”dysregulation of glucose metabolism contributes to…….various disease, such as cancer….”. This is not an accurate description. Especially in cancer, the tumor growth require large amount of ATP and nutrients, causes the switch to the fast ATP-generating process, that is Glycolysis. So, it is not surprising, if the aortic wall expansion, leading to the changes in glycolysis. But it should be discussed more in the text. Maybe this is the reason why restricting glycolysis improve AAA development.

Additional comments

Mistakes: Figure 4 and Figure 5 figures were not consistent with the text in the manuscript. Figure5 refers GO, KEGG analysis. Figure 4 were String analysis. Please correct it.

---

## Round 0.2 · accepted · Accept

The authors have addressed properly all the revisions requested.

·

Basic reporting

Authors have successfully responded to my comments.

Experimental design

None.

Validity of the findings

None.

·

Basic reporting

The authors answered my questions very carefully by supplying with more information about the secreted protein analysis in aged-related AAA formation. And authors had corrected their mistakes for the order of the figures.

Experimental design

None

Validity of the findings

Thanks to authors for offering more interesting data on secreted proteins. It is very interesting data from Table 2 and Figure 1B, indicating the age alone causes dramatic increases in several HuB proteins of DEPs, while four gens of them are secreted proteins. In mice, Vwf and Prdx2 may come from liver, which suggests the inter-organ communication from the other organs to regulate the AAA formation in old mice. Interesting data in terms of physiological significance.

Additional comments

Please make sure to upload good quality of figures.